# Loss of cohesin regulator PDS5A reveals repressive role of Polycomb loops

Daniel Bsteh [1,2,3,4], Hagar F. Moussa [1,2,5], Georg Michlits [1,2,6], Ramesh Yelagandula [1,7], Jingkui Wang[1,8], Ulrich Elling [1] & Oliver Bell [1,3] ✉

Polycomb Repressive Complexes 1 and 2 (PRC1, PRC2) are conserved epigenetic regulators that promote transcriptional gene silencing. PRC1 and PRC2 converge on shared targets, catalyzing repressive histone modifications. Additionally, a subset of PRC1/PRC2 targets engage in long-range interactions whose functions in gene silencing are poorly understood. Using a CRISPR screen in mouse embryonic stem cells, we found that the cohesin regulator PDS5A links transcriptional silencing by Polycomb and 3D genome organization. PDS5A deletion impairs cohesin unloading and results in derepression of a subset of endogenous PRC1/PRC2 target genes. Importantly, derepression is not linked to loss of Polycomb chromatin domains. Instead, PDS5A removal causes aberrant cohesin activity leading to ectopic insulation sites, which disrupt the formation of ultra-long Polycomb loops. We show that these loops are important for robust silencing at a subset of PRC1/PRC2 target genes and that maintenance of cohesin-dependent genome architecture is critical for Polycomb regulation.

In metazoans, precise epigenetic regulation of gene expression enables the development of diverse cell types despite the same underlying genomic blueprint. Gene expression is primarily controlled by DNA binding transcription factors directing the transcriptional apparatus. However, epigenetic mechanisms modulate chromatin to directly and indirectly regulate transcription. Polycomb repressive complexes are chromatin-modifiers and serve as prototypes of epigenetic gene regulation via histone modifications. Decades of research have cemented their roles in establishing and maintaining cell identity throughout development across organisms, ranging from *Drosophila melanogaster* to vertebrae[1,2]. Moreover, aberrant activity of Polycomb repressive complexes and other epigenetic regulators contribute to diverse diseases, including cancer initiation and metastasis,

highlighting the importance of understanding the pathological mechanisms[3–10].

Polycomb group (PcG) proteins are conventionally grouped into Polycomb repressive complexes 1 and 2 (PRC1 and PRC2). All PRC1 complexes share RING1A/B as their catalytic core subunit, which deposits monoubiquitination at lysine 119 of histone H2A (H2Aub1) at its targets, whereas PRC2 contains the histone methyltransferase EZH1/2, which catalyzes mono-, di- and trimethylation at lysine 27 of histone H3 (H3K27me1/2/3) at targets[11–14]. In vertebrates, PRC1 complexes have diversified into distinct subcomplexes based on incorporation of one of six paralogous PCGF proteins (PCGF1-6). PCGF2 or PCGF4 dictate assembly of canonical PRC1 (cPRC1) which specifically incorporates CBX (chromobox-containing protein) subunits. CBX

[1]Institute of Molecular Biotechnology of the Austrian Academy of Sciences (IMBA), Vienna BioCenter (VBC), Vienna, Austria. [2]Vienna BioCenter PhD Program, Doctoral School of the University of Vienna and Medical University of Vienna, Vienna, Austria. [3]Departments of Biochemistry and Molecular Medicine, and Stem Cell and Regenerative Medicine, Norris Comprehensive Cancer Center, Keck School of Medicine, University of Southern California, Los Angeles, CA, USA. [4]Present address: Division of Medical Oncology, Department of Medicine, Norris Comprehensive Cancer Center, Keck School of Medicine, University of Southern California, Los Angeles, CA, USA. [5]Present address: Department of Biomedical Engineering and Biological Design Center, Boston University, Boston, MA 02215, USA. [6]Present address: JLP Health GmbH, Himmelhofgasse 62, 1130 Vienna, Austria. [7]Present address: Laboratory of Epigenetics, Cell Fate & Disease, Centre for DNA Fingerprinting and Diagnostics (CDFD), Uppal, Hyderabad 500039, India. [8]Present address: Research Institute of Molecular Pathology (IMP), Vienna BioCenter (VBC), Vienna, Austria. ✉e-mail: oliver.bell@med.usc.edu

subunits endow cPRC1 with the capacity to bind H3K27me3, which promotes cPRC1 recruitment to PRC2 target genes and transcriptional silencing[11,15,16]. PCGF1, 3, 5, and 6 form variant PRC1 (vPRC1) complexes which harbor RING1 and YY1- binding protein (RYBP), or its paralogue YAF2 instead of CBX, and rely on H3K27me3-independent mechanisms of chromatin targeting. Thus, PRC1 and PRC2 are generally considered to exert their repressive functions in a synergistic manner, but they have different mechanisms of targeting, signaling, and repression (reviewed in[17]).

Recent studies have established that vPRC1 can act upstream of PRC2 and cPRC1, and that its deposition of H2Aub1 is critical for Polycomb-dependent gene silencing[18-20]. Indeed, loss of PRC2 or cPRC1 does not substantially compromise the repression of Polycomb target genes in mouse embryonic stem cells (ESCs) expressing vPRC1[18,19]. These findings have propelled vPRC1 to the center of coordinating and establishing a repressive Polycomb chromatin domain.

Although cPRC1 contributes minimally to H2Aub1 deposition, it possesses the unique capacity to mediate long-range 3D interactions between Polycomb target genes[12,21-27], which has been shown to contribute to gene silencing in flies[28].

The redundant functions of vPRC1 and cPRC1 complicate dissecting their individual mechanisms by genetic analysis[18,19,29,30]. To circumvent this limitation, we previously developed a Polycomb in vivo Assay that reports the activity of distinct PRC1 complexes. Briefly, we generated ESCs that can recruit ectopic cPRC1 or vPRC1 to an integrated TetO repeat sequence flanked by fluorescent reporters (Fig. 1a)[31]. For instance, ectopic expression of a CBX7-Tet repressor domain (TetR-CBX7) fusion triggers the assembly of cPRC1 at the TetO sites, Polycomb-dependent histone modifications and reporter gene silencing. Binding of the TetR fusion is released upon addition of Doxycycline (Dox), and we found that more than 70% of cells maintained cPRC1-induced, but not vPRC1-induced, silencing in the

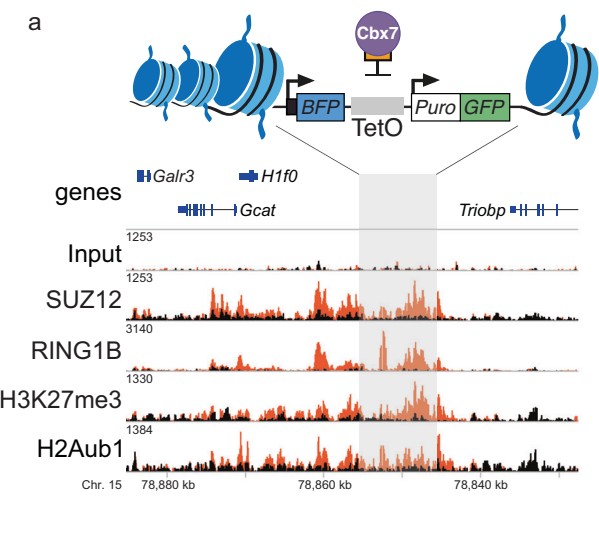

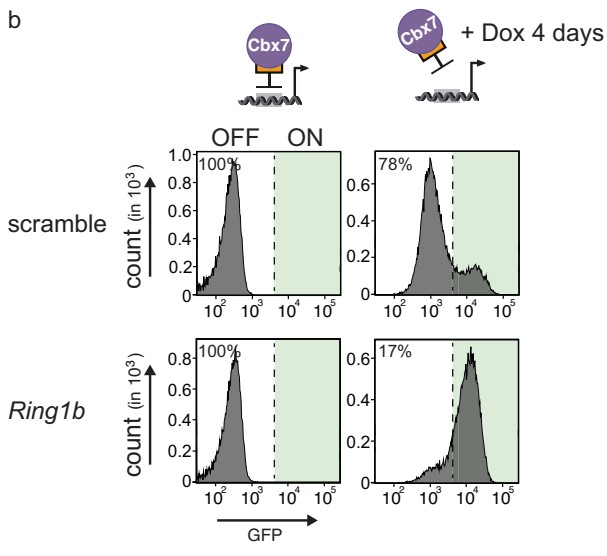

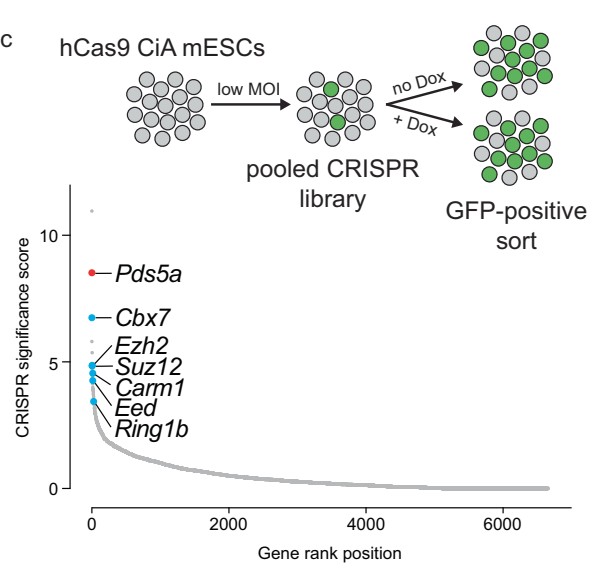

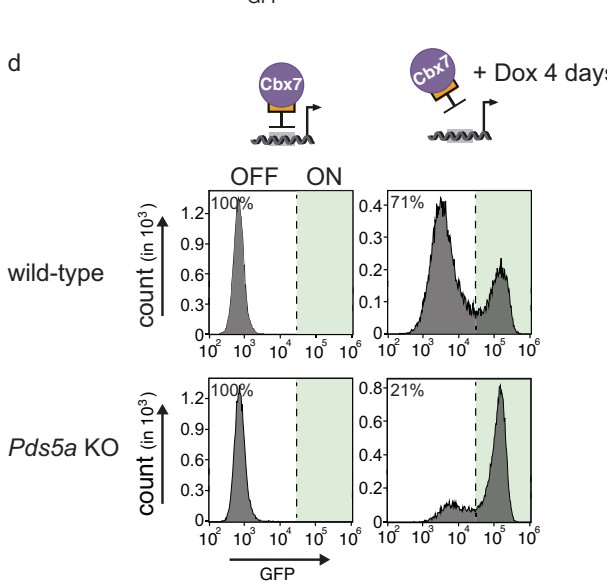

**Fig. 1 | CRISPR screen of cPRC1-dependent gene silencing reveals *Pds5a*.**
**a** Schematic of ectopic dual reporter locus consisting of 7x TetO landing sites flanked by an upstream Ef1a promoter driven BFP and a downstream PGK driven puromycin/GFP (top). Genomic ChIP-CapSeq screenshot of PcG proteins and histone modifications before (black) and after ectopic TetR-Cbx7 expression (orange) (bottom). Also see Supplementary Fig. 1a and Supplementary Table 3. **b** Flow cytometry histograms of GFP signal before (left) and after 4 days of Dox-dependent reversal of TetR-CBX7 tethering (right) in control (top) and *Ring1b* KO (bottom)

Polycomb reporter ESCs. Percentages refer to fraction of GFP-negative ESCs.
**c** Schematic of CRISPR screen design. MOI = multiplicity of infection. Genes (sgRNA library described in ref. 32) are rank-ordered based on CRISPR significance score (−log 10 MAGeCK significance score[32]; *n* = mean of three independent experiments). **d** Flow cytometry histograms of GFP signal before (left) and after 4 days of Dox-dependent reversal of TetR-CBX7 tethering (right) in control (top) and *PdsSa* KO (bottom) Polycomb reporter ESCs. Percentages refer to fraction of GFP-negative ESCs.

presence of Dox. We showed that sequence-independent propagation of cPRC1-induced silencing requires H3K27me3 and H2Aub1, suggesting that it relies on PRC1/PRC2 feedback. However, the mechanism and players required for heritable PRC1/PRC2-mediated silencing, and whether they are the same at all target genes, remain incompletely understood.

Here we performed a CRISPR-mutagenesis screen to identify novel regulators of silencing by PRC1/PRC2. We discovered that the cohesin regulator PDS5A is required for repression of canonical Polycomb target genes. Unexpectedly, loss of PDS5A does not substantially impact repressive Polycomb chromatin domains, but instead disrupts ultra-long chromatin loops between PRC1/PRC2 target genes. Our work uncovers a subset of Polycomb target genes that require distal 3D interactions for transcriptional silencing.

## Results

### CRISPR screen of cPRC1-induced gene silencing reveals *Pds5a* dependence

To identify novel regulators of PRC1/PRC2-mediated target gene silencing, we took advantage of the Polycomb in vivo Assay to perform CRISPR-based screening. Chromatin immunoprecipitation coupled with cost-efficient oligo capture sequencing (ChIP-CapSeq) confirmed that ectopic CBX7 tethering nucleated a Polycomb chromatin domain with high levels of RING1B, SUZ12, H3K27me3, and H2Aub1 surrounding the TetO nucleation site whereas reference loci including selected Polycomb and non-Polycomb target genes were unaffected (Fig. 1a, Supplementary Fig. 1a, Supplementary Table 3). To establish the screening platform, we introduced stable expression of hCas9 into our TetR-CBX7 reporter line. As a proof-of-principle, we infected this line with lentiviral vectors expressing either scramble sgRNA or sgRNA specific for *Ring1b*. We found that CRISPR mutation of *Ring1b* had a negligible effect on silencing induced by TetR-CBX7 (in the absence of Dox), but strongly impaired the epigenetic maintenance of silencing (in the presence of Dox, Fig. 1b), consistent with our previous observations[31]. Thus, our TetR-CBX7 reporter ESCs recapitulate epigenetic Polycomb-dependent gene silencing and are sensitive to genetic perturbations.

Using this platform, we performed pooled CRISPR screens with unique molecular identifiers (UMIs), which allow analysis of mutant phenotypes at a single-cell level[32]. The UMI CRISPR library contained approx. 27,000 sgRNAs targeting all annotated mouse nuclear protein-coding genes with four sgRNAs per gene. Each sgRNA was paired with thousands of barcodes representing UMIs, improving the signal-to-noise ratio and hit calling. hCas9-expressing TetR-CBX7 reporter ESCs were transduced with the pooled library and selected with neomycin. We used FACS to isolate GFP-positive cells (Fig. 1c and Supplementary Fig. 1b), and the unsorted population served as background control. Because GFP activation occurs at a very low frequency, we performed repeated FACS in the screen to enrich for GFP-positive cells. Relative enrichment of sgRNAs was determined by sequencing of UMIs in both populations followed by statistical analysis using MAGeCK[33].

We performed a screen with reporter cells cultured without Dox (Fig. 1c) and uncovered 51 genes that were significantly enriched in the GFP-positive cell population (*p*-value < 0.005) (Supplementary Table 1). We also performed a separate screen of cells treated with Dox for 3 days, but spontaneous GFP re-activation in some cells, independently of any mutation, precluded us from identifying statistically significant hits.

The top hits in our screen included genes encoding subunits of cPRC1 (*Cbx7, Ring1b*) and PRC2 (*Ezh2, Suz12, Eed*), as well as a negative regulator (*Carm1*) of the Polycomb repressive deubiquitinase complex (PR-DUB)[34]. This indicates that our screening approach identified known genes required for TetR-CBX7-induced silencing (*Cbx7*) and its epigenetic maintenance (*Ring1b, Suz12, Carm1*) (Fig. 1c). Notably, *Pds5a*, which encodes a regulator of the cohesin complex, was the

second most-significant hit in the screen. To validate this hit, we used CRISPR-Cas9 to target *Pds5a* independently, and observed reduced silencing in TetR-CBX7 reporter ESCs treated with Dox (Fig. 1d and Supplementary Fig. 1c). Thus, similar to *Ring1b*, *Pds5a* is required for the epigenetic maintenance of silencing induced by cPRC1.

The cohesin protein complex is composed of three core subunits, SMC1, SMC3 and RAD21 (also known as SCC1), which form a tripartite ring structure that entraps DNA[35]. Several auxiliary cohesin proteins are critical for dynamic regulation of DNA interactions. For instance, cohesin-dependent extrusion of chromatin loops involves STAG1/2, WAPL and PDS5A/B which associate at the interface between SMC3 and RAD21 and control ATPase activity and/or ring opening[36–45]. In addition to *Pds5a*, our CRISPR screen revealed enrichment of *Stag2*, albeit below the significance cutoff (*p*-value = 0.045). Since cohesin function is essential for cell division it is possible that other auxiliary cohesin proteins have been missed in our screen[46–48]. To separately evaluate the requirement of other cohesin release factors for Polycomb-dependent silencing, we used independent CRISPR-Cas9 mutagenesis targeting *Pds5b*, *Wapl*, *Stag1* and *Stag2* in TetR-CBX7 reporter ESCs and observed that similar to *Pds5a* maintenance of GFP silencing was impaired (Supplementary Fig. 1d). Since immunoblot detection of partial depletion would be challenging in the CRISPR mutant populations, we independently generated *Pds5b* knockout ESCs and observed reduced maintenance of reporter gene silencing, consistent with the CRISPR population experiment (Supplementary Fig. 1e, f). Together, these results suggests that regulation of cohesin activity by PDS5A and other auxiliary factors is critical for maintenance of Polycomb-dependent silencing.

### Loss of PDS5A results in de-repression of endogenous PRC1/PRC2 target genes

To determine the impact of PDS5A deletion on endogenous Polycomb-dependent gene regulation, we generated *Pds5a* knockout ESCs using CRISPR-Cas9 (*Pds5a* KO). In addition, we obtained a loss-of-function (LOF) ESC line harboring a disruptive gene-trap in the second intron of the *Pds5a* gene (*Pds5a*[GT] KO)[49]. Since gene-trap disruption is reversible, we also generated a matched control ESC line in which *Pds5a* expression was restored (*Pds5a*[GT] WT). *Pds5a* knockouts as well as the rescue were confirmed by western blot (Fig. 2a). PDS5A deletion did not impact the abundance of PDS5B, the cohesin subunit SMC3, the PcG proteins SUZ12 and RING1B, nor the global levels of their associated histone modifications (Fig. 2a and Supplementary Fig. 2a).

Given cohesin's essential role in sister chromatid cohesion, deletion of cohesin subunits frequently impairs cell proliferation, hampering the analysis of its precise function in gene regulation[46–48]. Although most cohesin proteins are essential for ESC viability[46–48], both PDS5A and STAG2 have paralogs with redundant but not identical functions that are sufficient to maintain self-renewal and proliferation[45,50]. Indeed, *Pds5a* KO ESC lines displayed characteristically dense colonies that could be stably maintained in culture, similar to wild-type ESCs (Supplementary Fig. 2b). Consistently, cell cycle profiles and pluripotency marker expression were highly comparable between KO and control ESCs, suggesting that PDS5A is largely dispensable for ESC self-renewal and proliferation (Supplementary Fig. 2b–d).

To evaluate how endogenous Polycomb target genes are affected by PDS5A deletion, we performed transcriptome profiling of *Pds5a*[GT] KO ESCs and *Pds5a* KO ESCs which revealed differential expression of 1029 and 1568 genes (DEGs), respectively (cutoff: adjusted *P* value ≤ 0.05; LFC ≥ 0.5) (Fig. 2b). DEGs were strongly correlated between the two mutant ESC lines (R = 0.77, p < 2.2e$^{-16}$) (Supplementary Fig. 2e). To categorize DEGs, we annotated transcription start sites (TSSs) in ESCs based on PRC1 and PRC2 occupancy, and enrichment of Polycomb-associated histone modifications (Supplementary Fig. 2f). Because non-methylated CpG-islands have emerged as a general feature of

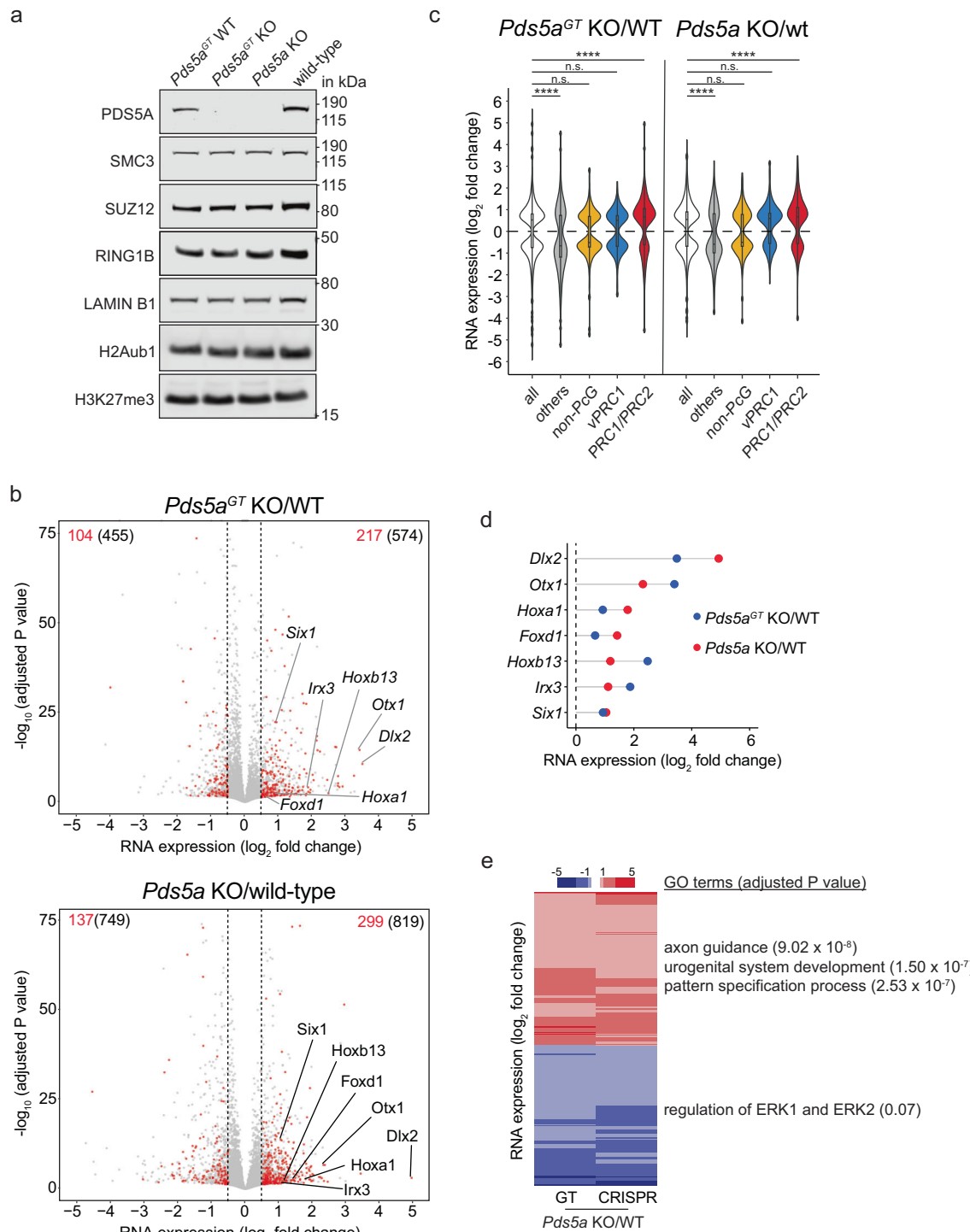

**Fig. 2 | Loss of cohesin regulator PDS5A results in de-repression of canonical PRC1/PRC2 target genes. a** Western blot of cohesin, PcG proteins and histone modifications in *Pds5a^GT* KO and *Pds5a^GT* WT ESCs and in *Pds5a* KO and wild-type ESCs. **b** Volcano plots show gene expression changes in *Pds5a^GT* KO vs. *Pds5a^GT* WT ESCs (top) and *Pds5a* KO vs. wild-type ESCs (bottom) (*n* = three replicates). Number of significantly up- or downregulated genes (black) (adjusted *P* value ≤ 0.05; LFC ≥ 0.5). Number of significantly up- and downregulated PRC1/PRC2 target genes (red). **c** Violin plots compare expression changes of all DEGs with different gene classes in *Pds5a^GT* KO ESCs and *Pds5a* KO ESCs. In the middle of each density curve is

a small box plot, with the rectangle showing the ends of the first and third quartiles and central dot the median. Significance was determined by Wilcoxon rank-sum test. Asterisks indicate significant differences between groups (**** $p < 1 \times 10^{-5}$; n.s. – not significant). **d** Dot plots show expression changes of selected PRC1/PRC2 target genes in *Pds5a^GT* KO ESCs and *Pds5a* KO ESCs. **e** Heatmap shows cluster analysis of common upregulated (red) and downregulated (blue) DEGs in *Pds5a^GT* KO ESCs and *Pds5a* KO ESCs. Most enriched gene ontology (GO) terms and corresponding significance are indicated for each cluster (right).

Polycomb target genes[51–56], we used available BioCap data to distinguish NMI TSSs from methylated, CpG-poor TSSs which we annotated as "others"[57]. We then classified NMI TSSs with overlapping peaks of RING1B and SUZ12 as "PRC1/PRC2 target genes", whereas NMI TSSs with RING1B peaks only were classified as "vPRC1 target genes". PRC1/PRC2 target genes displayed high levels of H3K27me3 and were predominantly bound by CBX7-containing cPRC1. In comparison, CBX7 was low at vPRC1 target genes which showed substantial PCGF1 occupancy instead. NMI TSSs lacking both RING1B (PRC1) and/or SUZ12 (PRC2) peaks within 3 kb of their TSSs were classified as "non-PcG target genes".

Based on this classification, our transcriptome analysis revealed that PRC1/PRC2 target gene expression was significantly increased in both $Pds5a^{GT}$ KO ESCs and $Pds5a$ KO ESCs (Fig. 2c, d), in agreement with a recent report[50]. Consistent with derepression of PRC1/PRC2 target genes, Gene Ontology (GO) analysis of upregulated genes revealed terms related to developmental processes such as neurogenesis and pattern specification (Fig. 2e). To determine if defects in PRC1/PRC2 target gene silencing are directly coupled to PDS5A loss, we used $Pds5a^{GT}$ WT ESCs enabling inducible LOF by reverting the gene-trap cassette into a disruptive orientation using FlpO recombinase. RT-qPCR analysis of selected PRC1/PRC2 target genes showed transcriptional upregulation within 72 h in FlpO-transduced $Pds5a^{GT}$ WT ESCs compared to mock control (Supplementary Fig. 2g).

Together, these results reveal that PDS5A is required for transcriptional silencing of endogenous PRC1/PRC2 target genes, validating our CRISPR screening results.

## PDS5A deletion has minimal effect on Polycomb chromatin domains

Polycomb-mediated transcriptional silencing is linked to the formation of repressive chromatin domains marked by PRC1 and PRC2 occupancy and deposition of Polycomb-dependent histone modifications[2,17]. We considered that impaired silencing of endogenous PRC1/PRC2 target genes upon PDS5A deletion results from erosion of these repressive chromatin domains. To test this hypothesis, we used calibrated ChIP-seq (cChIP-seq) of PcG proteins and Polycomb-dependent histone modifications in matching gene-trap ESCs to limit the influence of potential secondary changes due to long-term cohesin deregulation. Specifically, we analyzed enrichments of RING1B, SUZ12, H3K27me3 and H2Aub1 in $Pds5a^{GT}$ WT and $Pds5a^{GT}$ KO ESCs and validated the results with ChIP-qPCR of wild-type and $Pds5a$ KO ESCs (Fig. 3a–c, and Supplementary Fig. 3a–c). PDS5A deletion led to a modest reduction in RING1B and H3K27me3 signals at PRC1/PRC2 and vPRC1 target genes, but no differences in SUZ12 and H2Aub1 enrichment across the different gene classes. RING1B and H3K27me3 levels were uniformly reduced at PRC1/PRC2 and vPRC1 target genes independent of expression changes. In addition, we used ATAC-seq to analyze $Pds5a^{GT}$ WT and $Pds5a^{GT}$ KO ESCs but did not observe substantial differences in DNA accessibility, arguing that the modest difference in histone modifications does not affect the integrity of Polycomb chromatin domains (Fig. 3d and Supplementary Fig. 3d).

Together, these results show that PDS5A deletion has limited impact on the maintenance of Polycomb chromatin domains. Given the redundant activities of PRC1 and PRC2 in maintaining repression[19,29,30], the minimal reduction of PcG protein binding and histone modifications appears uncoupled from transcriptional activation at PRC1/PRC2 target genes upon loss of PDS5A, suggesting that $Pds5a$ KO impairs other mechanisms that are critical for transcriptional silencing.

## PDS5A colocalizes with cohesin and destabilizes chromatin binding

Recent evidence suggests that PDS5A functions to restrict chromatin loop extrusion at CTCF sites by binding to acetylated cohesin and

inhibiting its ATPase activity[45]. We used cChIP-seq to determine the genomic distribution of PDS5A relative to RAD21, CTCF and PcG proteins in $Pds5a^{GT}$ WT ESCs. As expected, PDS5A showed extensive overlap with RAD21 and CTCF (Supplementary Fig. 4a–d)[50,58,59]. In contrast, PDS5A was absent from Polycomb target genes, suggesting that PDS5A indirectly promotes silencing of PRC1/PRC2 target genes (Supplementary Fig. 4e).

Deletion of PDS5A alone or in combination with PDS5B has been shown to increase cohesin residence time on chromatin, resulting in continued loop extrusion, formation of larger TADs and loss of compartmentalization[43,45]. In ESCs, $Pds5a$ is considerably higher expressed than $Pds5b$ suggesting that it encodes the dominant protein paralog (Supplementary Fig. 4f). To evaluate potential differences in cohesin binding between $Pds5a^{GT}$ KO and $Pds5a^{GT}$ WT ESCs, we performed RAD21 cChIP-seq (Supplementary Fig. 4a–d).

Differential enrichment analysis of RAD21 peaks revealed a significant increase in occupancy at ~25% of targets (cluster 2) in $Pds5a^{GT}$ KO ESCs. Notably, using ChromHMM annotations[60], we found that compared to clusters 1 and 3, sites with strongly increased RAD21 occupancy were enriched in gene regulatory elements and relatively depleted in CTCF binding sequences (Supplementary Fig. 4g). In contrast, orthogonal analysis of bulk cohesin binding through nuclear fractionation did not show any discernible differences in SMC3 signal on chromatin (Supplementary Fig. 4h). However, since immunodetection is known to have limited sensitivity compared to cChIP-seq[43], we conclude that PDS5A loss promotes aberrant cohesin activity leading to increased occupancy at a substantial portion of the genome. Together, our findings support the notion that PDS5A regulates cohesin activity likely by restricting chromatin loop extrusion in ESCs.

## PDS5A deletion causes aberrant cohesin activity and TAD boundary violations

Based on our findings above, we hypothesized that loss of PDS5A leads to a cohesin-dependent dysregulation of 3D genome architecture that compromises the silencing of endogenous, PRC1/PRC2 target genes. To investigate if PDS5A deletion alters the 3D genome architecture, we performed in situ Hi-C on wild-type and $Pds5a$ KO ESCs, excluding potential structural effects of the gene trap cassette. After quality control, we combined sequencing reads of Hi-C replicates amounting to a total of ~365 million valid unique $cis$-contacts per genotype (Supplementary Table 2). We observed decreased interaction frequencies in Knight-Ruiz (KR)[61] normalized Hi-C contact matrices resulting in reduced "checkerboard" patterns of alternating A and B compartments (Fig. 4a). Based on eigenvector analysis, compartment signal was reduced, but compartment did not switch from A to B or vice versa (Fig. 4b and Supplementary Fig. 4i). Reduced compartmentalization in $Pds5a$ KO ESCs was further confirmed by their lower compartment strength, a related benchmark measuring interactions within compartments (A/A or B/B) compared to between compartments (A/B or B/A) (Supplementary Fig. 4j). When examining relative contact probabilities (RCP) as a function of genomic distance, we found reduced compartmentalization, manifested by a decrease in very long long-range contacts (>5 Mb) in $Pds5a$ KO ESCs relative to wild-type (Fig. 4c). Contact probabilities in the relative short-range (50–500 kb) were also slightly reduced. In contrast, interactions in the mid- to long-range (500 kb–5 Mb) were increased in $Pds5a$ KO ESCs relative to wild-type.

Whereas PDS5A deletion reduced compartmentalization, topologically associating domain (TAD) sizes were on average larger in $Pds5a$ KO ESCs compared to wild-type ESCs (Supplementary Fig. 4k). To explore differences in TAD structure and loop formation between wild-type and $Pds5a$ KO ESCs, we utilized publicly available high-resolution ESC Hi-C data to identify high-confidence TAD intervals and loops[62]. Aggregate TAD and loop analysis showed that PDS5A

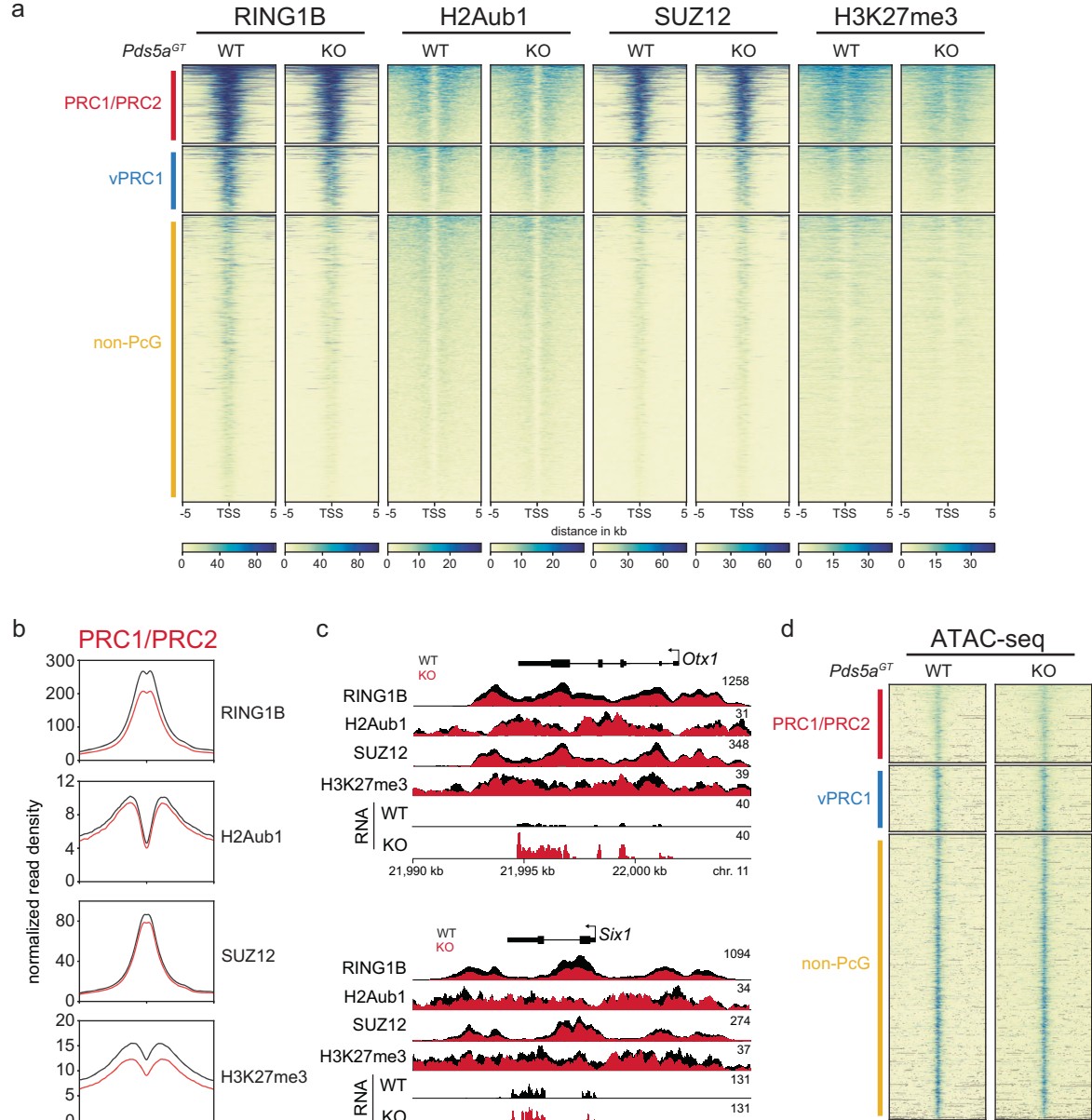

**Fig. 3 | PDS5A deletion has minimal effect on Polycomb chromatin domains.**
**a** cChIP-seq heatmaps of RING1B, H2Aub1, SUZ12 and H3K27me3 at PRC1/PRC2 target genes in *PdsSa^{GT}* WT and *PdsSa^{GT}* KO ESCs. Enrichment signal is plotted around the TSS (±5 kb) and clustered based on gene class annotation: PRC1/PRC2 target genes (red; *n* = 2895), vPRC1 target genes (blue; *n* = 2448), non-PcG genes (yellow; *n* = 10591). **b** Meta plots show average RING1B, H2Aub1, SUZ12 and H3K27me3 cChIP-seq signals in *PdsSa^{GT}* WT and *PdsSa^{GT}* KO ESCs. For each plot, normalized read density is plotted in 10 kb window centered around the TSS. **c** Genomic screenshot of cChIP-seq and RNA-seq in *PdsSa^{GT}* WT (black) and *PdsSa^{GT}* KO (red) ESCs. **d** ATAC-seq heatmaps in *PdsSa^{GT}* WT and *PdsSa^{GT}* KO ESCs. ATAC signal is plotted around the TSS (±5 kb) and clustered based on gene class annotation: PRC1/PRC2 target genes, vPRC1 target genes, non-PcG genes.

loss resulted in a relative contact reduction at pre-existing wild-type TADs and loops (Fig. 4d, e). We also detected increased interaction frequencies with neighboring TADs in *Pds5a* KO ESCs, whereas intra-TAD interactions were slightly decreased (Fig. 4f). These changes resemble those observed upon WAPL and/or PDS5A deletion in cancer cells, where increased cohesin residence time leads to an extension of chromatin loops, resulting in a genome-wide shift towards longer range interactions and violation of TAD boundaries[38,43,45]. Overall, our data suggest that PDS5A loss promotes aberrant cohesin loop extrusion in ESCs.

## PDS5A is required to maintain a subset of Polycomb loops
To explore how loss of PDS5A affects long-range interactions between Polycomb target genes, we used RING1B cChIP-seq to identify 525 Polycomb loops in wild-type ESCs (Fig. 5a and Supplementary Fig. 5a). This number is similar to 336 persistent interactions identified in cohesin-depleted ESCs[26], suggesting that Polycomb-associated long-range interaction account only for a small fraction of the 12,425 loops detected in ESCs. Further classification of Polycomb loops based on additional PcG proteins and associated histone modifications revealed that virtually all of them (476/525, 91%) arise from long-range

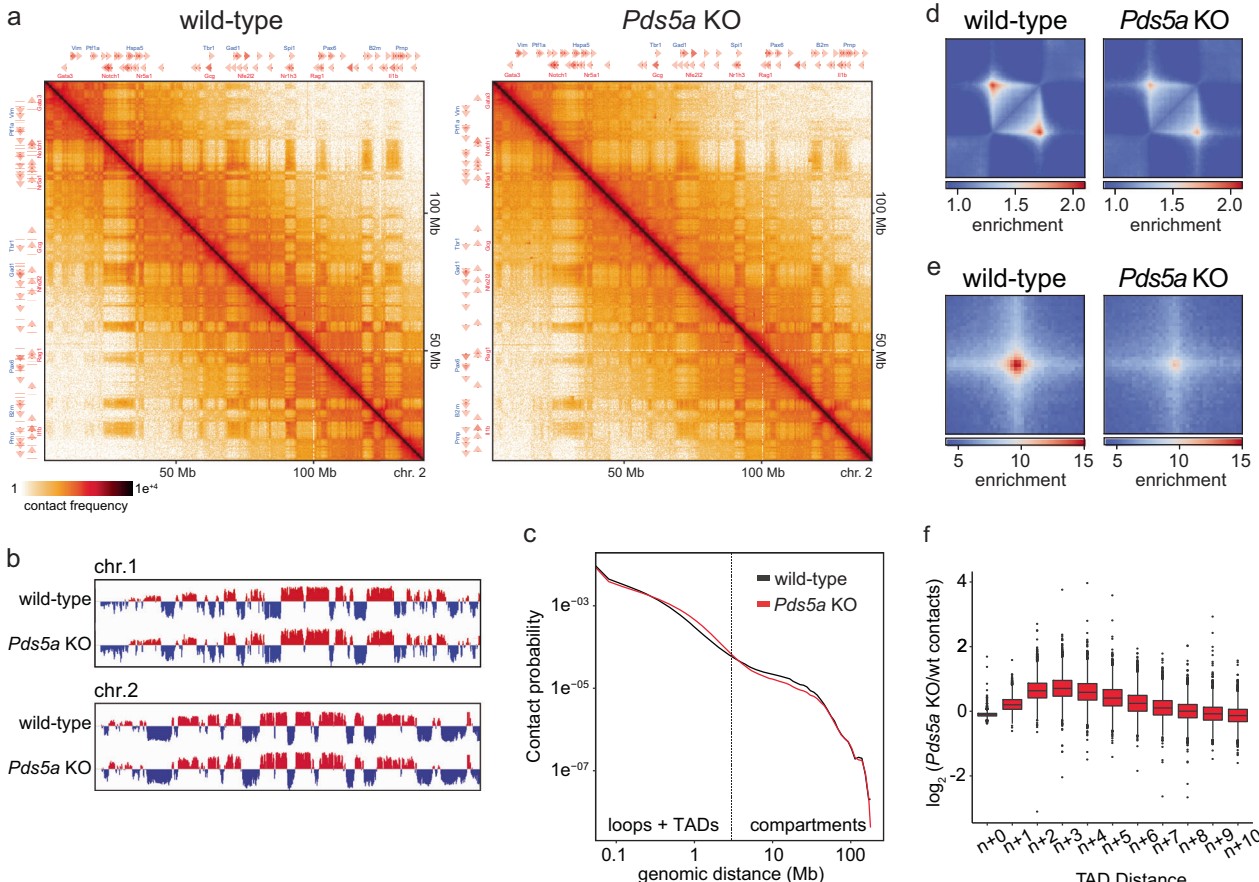

**Fig. 4 | PDS5A deletion causes aberrant cohesin activity and TAD boundary violations. a** Hi-C contact matrices of chromosome 2 in wild-type (left) and *Pds5a* KO (right) ESCs. **b** Eigenvector compartment signal tracks of chromosomes 1 and 2 comparing wild-type and *Pds5a* KO ESCs. **c** Genomic-distance-dependent contact probability from Hi-C in wild-type and *Pds5a* KO ESCs. Dashed line separates expected size ranges of TADs and compartments. **d** Aggregate TAD analysis in wild-type (left) and *Pds5a* KO (right) ESCs at 5-kb resolution. Effective contact probability is displayed using a set of published TADs in ESCs[62]. **e** Aggregate interaction analysis in wild-type (left) and *Pds5a* KO (right) ESCs at 5-kb resolution. Effective contact probability is displayed using a set of published loops in ESCs[62]. **f** Boxplot quantifies TAD boundary violation displaying changes in Hi-C contacts with n+ (x-axis) neighboring TADs in wild-type and *Pds5a* KO ESCs. Shown are median (horizontal line in the middle) and 25th to 75th percentiles (at the end of the boxes).

interactions between genomic sites harboring PRC1/PRC2 target genes, consistent with recent findings[26] (Fig. 5a and Supplementary Fig. 5a).

Unlike non-Polycomb (non-PcG) loops, which are substantially reduced in *Pds5a* KO ESCs, long-range interactions between PRC1/PRC2 target genes displayed on average only minor changes (Fig. 5a). We considered that aberrant cohesin activity and violation of TAD boundaries that we observed in *Pds5a* KO ESCs could interfere with a subset of Polycomb-associated long-range interactions in a locus-specific manner. Thus, we bifurcated Polycomb loops based on interactions between anchor sites harboring upregulated ("up", 65) and unchanged/downregulated ("not up", 411) PRC1/PRC2 target genes. Interestingly, anchor sites in wild-type ESCs that involve upregulated PRC1/PRC2 target genes are engaged in stronger loops compared to the anchor sites that involve unchanged/downregulated genes (Fig. 5b and Supplementary Fig. 5b). Strikingly, upon PDS5A deletion, the interaction frequency at upregulated anchor sites was dramatically reduced, whereas interactions between unchanged/downregulated anchor sites were relatively unaffected, similar to the class average (Fig. 5b). These results suggest that local dysregulation of cohesin-mediated chromosome architecture interferes with Polycomb-associated long-range interactions at a subset of PRC1/PRC2 target genes, which could compromise gene silencing.

## Polycomb loops crossing ultra-long distances are sensitive to cohesin dysregulation

To understand what defines the subset of Polycomb loops that are vulnerable to cohesin dysregulation, we first compared chromatin modifications between anchor sites of upregulated and unchanged/downregulated PRC1/PRC2 target genes. We noticed that at Polycomb loops of upregulated PRC1/PRC2 target genes, only one of the two anchor sites was associated with loss of gene silencing. To investigate potential differences in the repressive chromatin modifications, we separated the two anchor sites into unchanged (left) and upregulated (right) and compared PcG protein occupancy and associated histone modifications (Fig. 5c and Supplementary Fig. 5c). Anchor sites of unchanged/downregulated PRC1/PRC2 target genes served as the control dataset. Surprisingly, despite differential expression we found that upregulated (right) and unchanged anchor sites (left) had comparable repressive chromatin domains with similar reduction in RING1B occupancy and H3K27me3 upon PDS5A deletion (Fig. 5c and Supplementary Fig. 5c, d). Chromatin modifications at upregulated anchor sites were also similar to those at anchor sites of unchanged/downregulated PRC1/PRC2 target genes. Together, these results corroborate our genome-wide analysis revealing minimal reduction of repressive chromatin modifications at PRC1/PRC2 target genes and strongly suggest that reduced long-range interactions and loss of gene

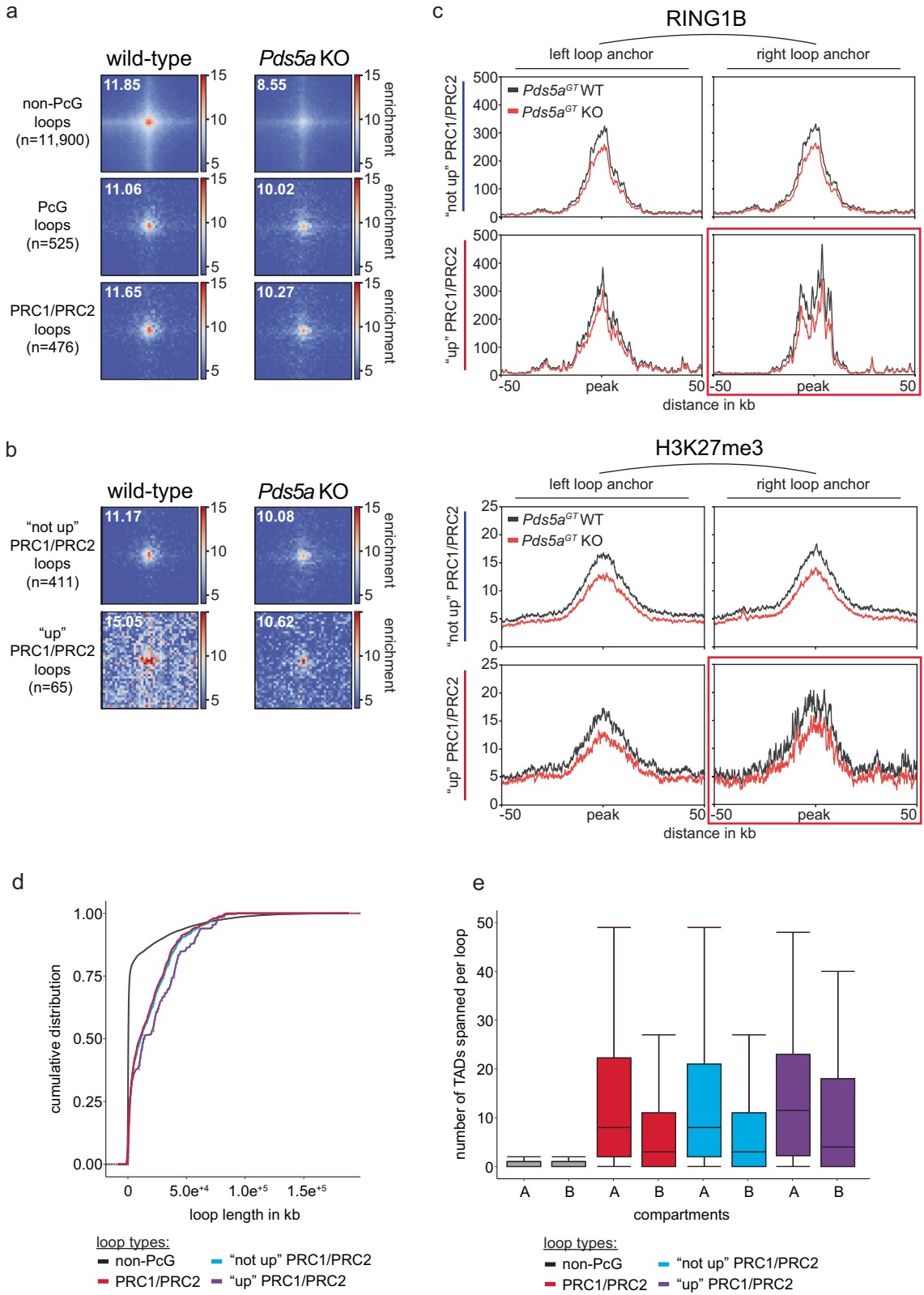

silencing in *Pds5a* KO ESCs are largely uncoupled from changes in Polycomb chromatin domains.

Next, we explored if sensitivity to cohesin dysregulation is linked to the distance between Polycomb loop anchor sites. Comparison of loop sizes of upregulated and unchanged/downregulated PRC1/PRC2 target genes revealed a striking difference: loops between anchor sites of upregulated PRC1/PRC2 target genes were substantially longer than loops between anchor sites of unchanged/downregulated or all shared PRC1/PRC2 target genes (Fig. 5d). Not surprisingly, this length bias was also reflected in a greater number of TADs within the A compartment traversed by Polycomb loops of upregulated PRC1/PRC2 target genes (Fig. 5e). Based on these results we conclude that ultra-long Polycomb loops are most vulnerable to cohesin dysregulation. We speculate that traversing a greater number of TADs increases the probability of

**Fig. 5 | PDS5A is required for maintenance of ultralong-range Polycomb loops.**
**a** Loop pileup analysis of non-PcG loops ($n = 11,900$), PcG loops (PRC1/2 and vPRC1; $n = 525$) and PRC1/PRC2 loops (PRC1/2 only; $n = 476$) in wild-type (left) and *Pds5a* KO (right) ESCs. Number indicates relative peak enrichment. **b** Loop pileup analysis of Polycomb loops overlapping unchanged/downregulated PRC1/PRC2 target genes (not up PRC1/PRC2 loop; $n = 411$) and upregulated PRC1/PRC2 target genes (up PRC1/PRC2 loop; $n = 65$) in wild-type (left) and *Pds5a* KO (right) ESCs. Number indicates relative peak enrichment. **c** Meta plots show average RING1B and H3K27me3 cChIP-seq signals at Polycomb loop anchors associated with unchanged/downregulated PRC1/PRC2 target genes (not up PRC1/PRC2 loop; $n = 411$) and upregulated PRC1/PRC2 target genes (up PRC1/PRC2 loop; $n = 65$) in wild-type (left) and *Pds5a* KO (right) ESCs. Red box indicates loop anchor that overlaps upregulated PRC1/PRC2 target genes in *Pds5a* KO ESCs. **d** Cumulative distribution of chromatin loop lengths. **e** Box plot shows number of TADs traversed by different types of chromatin loops in A or B compartments. Shown are median (horizontal line in the middle), 25th to 75th percentiles (at the end of the boxes) and 90% percentiles (whiskers).

interference by cohesin-mediated loop extrusion and TAD boundary violations. Importantly, by uncoupling loss of Polycomb loops from changes in repressive chromatin modifications, these results reveal a subset of PRC1/PRC2 target genes that potentially depends on long-range interactions for silencing.

### Loss of Polycomb loops is linked to cohesin-mediated insulation gain

To uncover the potential mechanism by which aberrant cohesin activity interferes with ultra-long Polycomb loops between PRC1/PRC2 target genes, we defined regions in the genome with significant local changes in 3D chromosome architecture. Specifically, we calculated insulation scores[63] in 250 kb bins across the genomes of wild-type and *Pds5a* KO ESCs. Insulation scores in most bins (40,838) were unchanged upon PDS5A deletion (Supplementary Fig. 6a, b). Additionally, we identified 17,269 bins with significant reduction in insulation in *Pds5a* KO ESCs, suggesting loss of TAD boundaries in response to cohesin dysregulation (Fig. 6a and Supplementary Fig. 6b).

Intriguingly, we identified 1406 genomic regions that gained insulation in *Pds5a* KO ESCs (Fig. 6a and Supplementary Fig. 6b). It is possible that the insulation gaining regions are directly linked to sites with increased cohesin occupancy (Supplementary Fig. 4b), e.g., stabilized loops at gene promoters and enhancers (cluster 2), but the spatial resolution for scoring insulation changes precludes a precise determination. In any case, we reasoned that newly formed insulation sites could interfere with ultra-long Polycomb loops between PRC1/PRC2 target genes. To explore this scenario, we analyzed the distances between newly formed insulation sites and Polycomb target genes. Strikingly, new insulation sites are located significantly closer to upregulated PRC1/PRC2 target genes than to all other classes of Polycomb and non-Polycomb genes (Fig. 6b). These data suggest that proximal changes in cohesin-mediated 3D chromosome architecture disrupt ultra-long Polycomb loops, potentially causing de-repression of a subset of PRC1/PRC2 target genes.

One prominent example of such cohesin-dependent dysregulation is the insulation gain region located between *Dlx2* and the *Hoxd* gene cluster (Fig. 6c). In wild-type ESCs, *Dlx2* forms strong interactions traversing ~3 Mb with the *Hoxd* gene cluster. Upon PDS5A deletion, these long-range interactions are lost and *Dlx2* expression is upregulated by more than 8-fold (Figs. 2b, d, and 6c), yet PcG protein occupancy and associated histone modifications are either unaffected or only marginally reduced at *Dlx2* and the *Hoxd* gene cluster. Instead, PDS5A deletion leads to a gain in insulation with increased cohesin binding near the *Sp9* gene, which is located between *Dlx2* and the *Hoxd* gene cluster. Virtual 4-C viewpoints from the insulation gaining region (v2), as well as from *Dlx2* (v1) and the *Hoxd* gene cluster (v3), reveal increased mid- to long-range interactions in *Pds5a* KO ESCs, consistent with aberrant extension of cohesin loops (Fig. 6c). This shift towards longer-range interactions is captured in Hi-C matrices as strengthening of two domains in between *Dlx2* and the *Hoxd* gene cluster (dashed triangles) (Fig. 6c). We speculate that aberrant extension of chromatin loops resulting from reduced cohesin unloading upon PDS5A loss strengthens ectopic cohesin occupancy and interaction domains at the expense of the ultra-long Polycomb loop between *Dlx2* and the *Hoxd* gene cluster. A similar example showcasing how insulation gain might interfere with Polycomb long-range interaction and gene repression is represented by contacts between *Hoxb13* and *Neurod2* (Supplementary Fig. 6c). Together, these results suggests that formation of ectopic insulation sites as result of aberrant cohesin activity interferes with maintenance of long-range interactions between PRC1/PRC2 target genes.

### Polycomb looping is required for *Foxd1* repression

Our results suggest that disruption of Polycomb loops is linked to aberrant expression of PRC1/PRC2 target genes. One possible explanation is that long-range Polycomb interactions between PRC1/PRC2 targets directly contribute to repression. Alternatively, extrusion of larger cohesin loops creates new TADs in which PRC1/PRC2 target genes are juxtaposed next to gene regulatory elements such as enhancers that are otherwise insulated from them.

To test the latter scenario, we analyzed the distances between H3K27ac peaks, marking active enhancers and promoters, and Polycomb target genes. We found that H3K27ac peaks are located at similar distances to unchanged/downregulated and to upregulated PRC1/PRC2 target genes, arguing against a causal link between loss of Polycomb repression and proximity to active enhancers and promoters (Extended Fig. 6d).

Next, we tested if loss of Polycomb long-range interactions by genetic deletion of a loop anchor region would impair PRC1/PRC2 target gene silencing comparable to PDS5A deletion. 65 PRC1/PRC2 target genes lost long-range interactions and displayed transcriptional upregulation upon PDS5A deletion. From this list, we selected *Foxd1* which interacts with *Irx2* across a distance of 25 Mb and is aberrantly activated in *Pds5a^GT* KO and *Pds5a* KO ESCs (Figs. 2d and 6d). We used CRISPR genome editing at *Irx2* to excise a 6.4 kb fragment overlapping SUZ12 and RING1B peaks in wild-type ESCs. After confirming sequence deletion by genotyping PCR, we isolated two independent ESC clones for RT-qPCR expression analysis (Fig. 6e and Supplementary Fig. 6e). Notably, both *Irx2* mutant clones showed transcriptional upregulation of *Foxd1* expression by >2.5-fold compared to wild-type ESCs. This effect was specific because expression of other PRC1/PRC2 target genes with unrelated Polycomb loops (*Ccno, Dll1, Barx1*) remained unchanged.

Taken together our results demonstrate that long-range interactions with *Irx2* are required for *Foxd1* silencing suggesting that Polycomb loops directly contribute to maintenance of repressive chromatin domains at a subset of PRC1/PRC2 target genes. Thus, Polycomb repression takes place in a delicate spatial equilibrium with cohesin-dependent nuclear architecture, that is essential to maintain robust silencing at PRC1/PRC2 target genes.

### Discussion

Here, we used a CRISPR-mutagenesis screen to identify novel regulators of cPRC1-induced gene silencing which revealed the cohesin regulator PDS5A. Subsequent independent deletion in ESCs confirmed that PDS5A, PDS5B, WAPL, STAG1 and STAG2 are genetic dependencies in reporter gene silencing and that PDS5A is required for repression of a subset of endogenous PRC1/PRC2 target genes. Notably, loss of PRC1/PRC2 silencing upon PDS5A deletion is mostly uncoupled from changes in Polycomb chromatin domains. Instead,

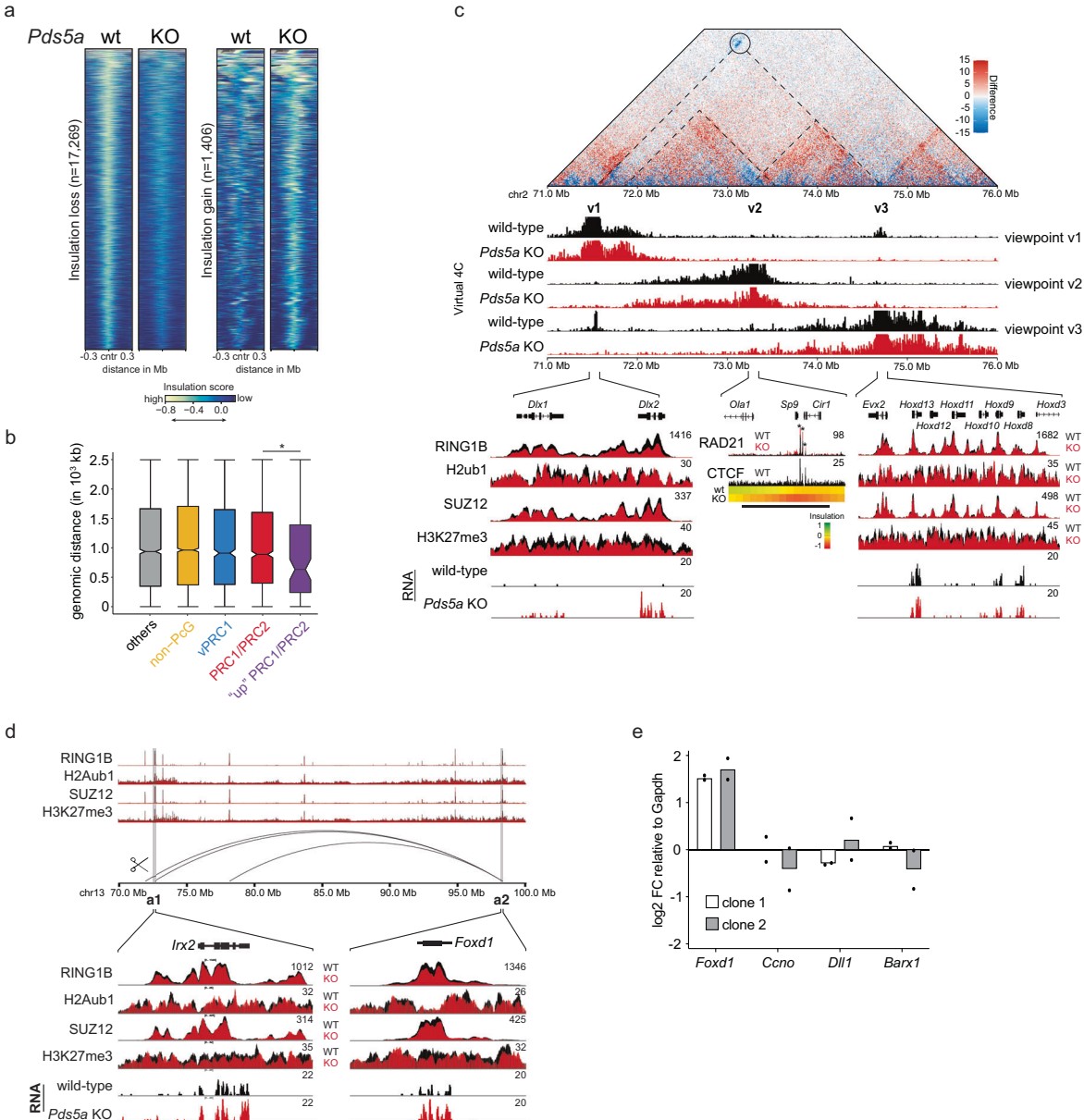

**Fig. 6 | Loss of Polycomb loops is linked to cohesin-mediated insulation gain.**
**a** Heatmaps show insulation scores in wild-type and *Pds5a* KO ESCs for differentially
insulated regions that lose (left, *n* = 17,269) or gain (right, *n* = 1,408) insulation upon
PDS5A deletion. Insulation scores are plotted ±300 kb around differentially insu-
lated regions. **b** Boxplots shows genomic distances of gene class TSSs to the closest
insulation-gaining regions. Shown are median (horizontal line in the middle), 25th
to 75th percentiles (at the end of the boxes) and 90% percentiles (whiskers). Sig-
nificance was determined by Wilcoxon rank-sum test. Asterisks indicate significant
differences between groups (* *P* value = 0.039). **c** Top: Hi-C matrix shows interac-
tion differences on chromosome 2 between wild-type and *Pds5a* KO ESCs (blue =
loss; red = gain). Middle: Virtual 4C contact plots compare interaction frequencies
at three viewpoints (v1, *Dlx1*, and *Dlx2*), (v2, insulation-gaining region indicated with
black bar) and (v3, *Hoxd* gene cluster) in wild-type (black) and *Pds5a* KO ESCs (red).

Bottom: Genomic screenshots of cChIP-seq of PcG proteins and histone mod-
ifications at v1 and v3, and of normalized RNA-seq counts in wild-type (black) and
*Pds5a* KO (red) ESCs. Genomic screenshot at v2 shows RAD21 cChIP-seq in wild-type
(black) and *Pds5a* KO (red) ESCs, CTCF ChIP-seq in wild-type (black), and insulation
score heatmaps. Asterisks indicate significantly increased RAD21 binding.
**d** Genomic screen shot showing cChIP-seq signals of PcG proteins and histone
modifications in wild-type (black) and *Pds5a^{GT}* KO (red) ESCs and Polycomb loops in
wild-type ESCs. *Irx2* (a1) and *Foxd1* (a2) are highlighted (gray) and cChIP-seq of PcG
proteins and histone modifications and RNA-seq of normalized expression counts
are shown at higher resolution below. Scissors indicate CRISPR excision at *Irx2*
locus. **e** RT-qPCR analysis of *Foxd1*, *Ccno*, *Dll1* and *Barx1* expression changes relative
to *Gapdh*. Shown are data of experimental replicates in two independent CRISPR
excision ESC clones. Source data are provided as a Source Data file.

PDS5A loss affects cohesin-dependent genome architecture. We
speculate that PDS5A deletion promotes aberrant chromatin loop
extrusion leading to breached TAD boundaries, increased cohesin
occupancy at gene regulatory elements and formation of ectopic
insulation sites. In turn, new insulation sites perturb competing loops
mediated by cPRC1. Finally, our results argue that ultra-long Polycomb
loops are critical for robust silencing at a subset of PRC1/PRC2
target genes.

Previous reports have linked genetic mutations in genes encoding
cohesin regulators to defects in Polycomb-dependent gene silencing,
but the underlying mechanisms remained unclear. For example, a
genetic screen for dominant suppressors of Polycomb-dependent
silencing in *Drosophila* revealed several mutants in the *wapl* gene[64]. In
mammalian cells, PDS5A and PDS5B are required for Polycomb-
dependent silencing[50], and *Pds5a* KO mice exhibit developmental
abnormalities, including skeletal malformations[65], that resemble the

patterning defects of Polycomb mutant mice[24,66,67]. Our results provide a mechanistic explanation linking changes in 3D genome architecture to defects in Polycomb-dependent gene silencing. While previous work demonstrated that cohesin activity counteracts Polycomb-mediated long-range interactions[26], we show that by disrupting Polycomb loops, PDS5A defects uncover the architectural role of PRC1 in gene silencing. Since loss of PRC1/PRC2 target gene silencing is uncoupled from changes in Polycomb chromatin domains, our data argue that Polycomb-dependent long-range interactions are required for and directly contribute to transcriptional repression.

The capacity of cPRC1 to form 3D chromatin interactions that contribute to gene silencing has been demonstrated in *Drosophila*[21,28]. To our knowledge, the contribution of cPRC1-mediated loops to gene silencing in mammalian cells has been controversial[22]. PRC1/PRC2 target genes are generally located within the active (A) compartment in the nucleus of ESCs and thereby in spatial proximity to actively transcribed genomic regions[68]. We speculate that Polycomb-dependent silencing of PRC1/PRC2 target genes involves multiple parallel mechanisms including repressive histone modifications and long-range interactions. By uncoupling loss of Polycomb silencing and loop interactions from changes in repressive chromatin modifications, our data argue that 3D organization by itself has repressive function potentially by tethering PRC1/PRC2 target genes away from transcriptional co-activators and/or the transcriptional machinery. Therefore, when combined with repressive chromatin modifications, which promote Polycomb feedback mechanisms, spatial aggregation by Polycomb loops would effectively enhance robust gene silencing.

The relative contribution of each of these mechanisms is likely locus-specific and may vary in different cell types. For example, alternative incorporation of paralogous cPRC1 subunits, such as CBX proteins or PHC proteins may influence the specific regulation of Polycomb 3D network formation as a mechanism of repression. Hence, future studies are needed to discern how cPRC1 and Polycomb-dependent genome architecture control target gene silencing in the context of different cell types and in disease.

## Methods

### Cell lines
All diploid ESC lines used in this study were derived from originally haploid HMSc2 termed AN3-12[49]. TetR-CBX7 reporter ESCs with 7x TetO DNA binding sites flanked by GFP and BFP reporter genes were previously described ref. 31. $Pds5a^{GT}$ KO and its corresponding wild-type ESCs with genetrap insertion in the non-disruptive orientation were acquired from the Haplobank repository (Cell IDs: 10388IH and 10388MH)[49].

### Cell culture conditions
All ESCs were cultivated without feeders in high-glucose-DMEM (Corning 10-013-CV) supplemented with 13.5% fetal bovine serum (Corning 35-015-CV), 10 mM HEPES pH 7.4 (Corning, 25-060-CI), 2 mM GlutaMAX (Gibco, 35050-061), 1 mM Sodium Pyruvate (Corning 25-000-CI), 1% Penicillin/Streptomycin (Sigma, P0781), 1X non-essential amino acids (Gibco, 11140-050), 50 mM β-mercaptoethanol (Gibco, 21985-023) and recombinant LIF. Cells were incubated at 37 °C and 5% $CO_2$ and were passaged every 48 h by trypsinization in 0.25% 1x Trypsin-EDTA (Gibco, 25200-056). In order to reverse of TetR-CBX7 fusion protein binding 1 µg/ml Doxycycline (Sigma, D9891) was added to cell culture medium.

### Generation of CRISPR-Cas9 mutants
Generation of mutant TetR-CBX7 ESC lines was achieved by CRISPR-Cas9 technology using a modified version of the vector plasmid pX330-U6-Chimeric_BB-CBh-hCas9 (Addgene #42230) that yields a BFP marker for selection (Gift by J. Zuber). Plasmids expressing hCas9 together with sgRNAs targeting *Pds5a* (5'-TGTCTCTGCAGA GTGGAACG- 3') or *Ring1b* (5'- GTGTTTACATCGGTTTTGCG -3') were transduced into parental reporter ESCs by electroporation using NEON transfection system (Invitrogen, MPK5000). 36 h post transfection cells were FACS sorted for hCas9-BFP and 1000-2000 cells seeded for clonal expansion on a 15 cm plate. 7–10 days later colony forming clones were individually picked, the targeted loci genotyped and loss of function confirmed by western blot.

For the generation of population mutants of TetR-CBX7 ESC cells we generated a stably expressing hCas9 clonal cell line that yields a hygromycin selection marker for hCas9. Further four guides (see Supplementary Table 4) for *Pds5b*, *Stag1*, *Stag2*, *Wapl*, each and a scramble control (5' GATCCATGTAATGCGTTCGA 3') were cloned in custom sgRNA vector including a G418 selection cassette and introduced into TetR-CBX7 hCas9 expressing mESCs via electroporation.

### CRISPR excision of Polycomb loop anchor
For excision of Polycomb loop anchors the mm10 genomic region chr13:72,626,687-72,633,437 was targeted. 2 sgRNAs 4,389 bp apart flanking the regional CpG island within that region were designed (5' GCTCTGAAGCTAGTAGAGGG 3', 5' CCTTCTGCGGTACAATACCG 3') and cloned into the CRISPR-Cas9 pX330-U6-Chimeric_BB-CBh-hCas9 (Addgene #42230) BFP-selection marker modified version described above (Gift by J. Zuber). The 2 sgRNA CRISPR-Cas9 plasmids were transduced wild-type mESCs via electroporation using NEON transfection system (Invitrogen, MPK5000). 36 h post transfection cells were FACS sorted for BFP and cells seeded for clonal expansion on a 15 cm plate. Single colony forming clones were isolated 7–10 days later and subjected to genotyping of the Irx2 Polycomb loop anchor site targeted for excision. Genotyping primers flanking the region as well as overlapping the borders of the excision target site were designed (see annotation and depiction in Supplementary Fig. 6e: ("a": CTCCAGTCC ATCACTACAATTG, "b": GCTAAGTTGGTCCAAAGGTC, "c": CCATAC CTGCTCCCTTTCCC, "d": GTCCCGGGCCTAGAAAATG).

### Hoechst-staining
For cell cycle profiling ESCs were trypsinized and genomic DNA was stained with Hoechst 33342 (20 mM; Thermo Fisher Scientific Cat. # 62249) for 30 min at 37 °C and 5% $CO_2$. Hoechst immunofluorescence was measured by flow cytometry on a FACSAria II cell sorter (BD Biosciences).

### AP-staining
One thousand cells were seeded and grown to form colonies at low density on 15 cm tissue culture dishes for 7 days. On day 7, dishes were washed with 100 mM tris (pH 8) and AP activity assay was performed using the VECTOR Blue AP Substrate Kit (Vector Laboratories, VECSK-5300) according to the manufacturer's instructions. Following AP staining, stained colonies were fixed in 4% formaldehyde overnight. Plates were rinsed with 1x PBS the following day and images taken on a brightfield microscope (EVOS XL Core system).

### Pluripotency marker staining
To assess pluripotency of PDS5a loss of function mutants, we applied intracellular staining of OCT3/4, SOX2 and SSEA1. Single cell suspensions of wild-type and Pds5a KO ESCs were permeabilized and fixed using the fixation/permeabilization buffer (R&D systems), washed twice with 1x PBS and stained using the H/MM pluripotent Stem Cell Multi-Color Flow Cytometry kit (R&D Systems) according to vendor's protocol. Flow cytometry data was collected on an Attune NxT equipped with Attune NxT v3.1 acquisition software. Final data analysis was performed using FlowJo (10.7.1).

## Western blot

10 million ESCs were subsequently lysed in Buffer A (25 mM Hepes pH 7.6, 5 mM MgCl$_2$, 25 mM KCl, 0.05 mM EDTA, 10% Glycerol, 1 mM DTT, 1 mM PMSF, 1× Complete Mini protease inhibitor, Roche) resuspended in RIPA buffer (150 mM NaCl, 1% triton, 0.5% sodium deoxy-cholate, 0.1% SDS, 50 mM Tris pH 8.0). Lysates were homogenized by sonication using a Bioruptor Pico (Diagenode) and concentration determined by Bradford assay (Biorad). 4x non-reducing Laemmli SDS sample buffer (Alfas Aesar, #J63615AD), 10 mM final DTT and 0.5% final BME were added to 20 μg total protein/sample and boiled at 95 °C for 5 min. Samples were separated on NuPAGE 4−12% Bis-Tris gels (Invitrogen) in Bis-Tris running buffer (Novues Biologicals) and transferred on a Merck Chemicals Immobilon-FL Membrane (PVDF 0.45 μm). After blocking the membranes (5% non-fat dry milk in 1× PBS, 0.1% Tween 20) the blots were incubated o/n with the primary antibodies in 5% non-fat dry milk in 1× PBS and 0.1% Tween 20. Antibodies used: PDS5a (Millipore Sigma #SAB2101764) 1:1000; PDS5B (Bethyl Laboratories A300-537A) 1:1000; RING1B (Cell Signaling D22F2) 1:1000; SMC3 (Bethyl Laboratories A300-060A) 1:2000; SUZ12 (Cell Signaling D39F6) 1:1000; LAMIN B1 (Abcam ab16048) 1:15000; H2AK119ub (Cell Signaling D27C4) 1:20000; H3K27me3 (Diagenode p069-050) 1:1000; H3 (Abcam ab1791) 1:10000; a-TUBULIN (Sigma-Aldrich T9026) 1:500. Next, membranes were incubated with corresponding secondary IRDye 800CW Goat anti-Rabbit IgG (H + L) (LICOR) or IRDye 680RD Goat anti-Mouse IgG (H + L) (LICOR) antibodies and imaged on an Odyssey CLx Near-Infrared Imaging System (LICOR).

## Nuclear protein fractionation assay

Nuclear soluble and chromatin bound fractions were isolated as described previously[69]. 10 million mESCs were collected and incubated at 4 °C in 1 ml Low-Salt Buffer (LSB) (20 mM HEPES (pH 7.9)), 1.5 mM MgCl$_2$, 25% glycerol, 2 mM EDTA, 1 mM DTT, 1 mM PMSF and protease inhibitor cocktail (Complete EDTA-free, Roche) for 15 min. Nuclei were isolated by adding NP-40 to a final concentration of 0.75% and gentle resuspension. After centrifugation (500 g for 5 min) the nuclear pelleted was then resuspended in 100 μl LSB. 100 μL of High-Salt Buffer (HSB) (25% glycol, 0.4 M NaCl, 20 mM HEPES pH 7.9, 1.5 mM MgCl$_2$, 1 mM DTT, 1 mM PMSF and protease inhibitor cocktail) (Complete EDTA-free, Roche) was added dropwise while vortexing at low speed to reach a final concentration of 200 mM NaCl. Further, samples were vortexed at low to medium speed at 4 °C for 30 min and centrifuged at 21,000 g for 10 min. The resulting supernatant was collected as the nuclear soluble fraction. The pellet containing the chromatin-bound protein fraction was washed twice in the 200 mM NaCl buffer, resuspended in SDS-PAGE loading buffer, sonicated using a probe sonicator and boiled at 95 °C for 10 min. Finally, both nuclear soluble and chromatin bound fractions were subjected to SDS-PAGE and western blot.

## Genetic CRISPR-Cas9 screen

For the genetic CRISPR-Cas9 mutagenesis screen, EF1a promoter driven hCas9 with a hygromycin resistance marker (modified version of Addgene #52961) was stably integrated via lentiviral transduction into the previously described TetR-CBX7 reporter ESC cell line, which contains 7x TetO DNA binding sites flanked by GFP and BFP reporter genes[31].

For CRISPR-Cas9 mutagenesis, a sgRNA library targeting 6560 nuclear factors with four sgRNAs per gene and 112 nontargeting controls was utilized[32]. For retroviral library generation, the barcoded plasmid library of sgRNAs, containing neomycin resistance for selection, was packaged in PlatinumE cells (Cell Biolabs) according to the manufacturer's recommendations.

$3 \times 10^8$ TetR-CBX7 reporter ESCs were infected with a 1:10 dilution of the harvested virus-containing supernatant PlatinumE cell medium for 24 h in the presence of 2 μg/ml polybrene (Santa Cruz

Biotechnology, SACSC-134220). The $3 \times 10^8$ ESCs were divided into three sets of $1 \times 10^8$ million ESCs (10 × 15 cm plates of 10 million cells each) that were treated as three separate replicates throughout entirety of the mutagenesis screen and sgRNA NGS sequencing. 24 h post infection, neomycin-resistance selection was started on the infected cells by addition of G418 (Gibco) at 0.5 mg/ml. After 24 h of selection, each replicate was expanded from 10 15-cm dishes to 20 dishes. Subsequently, for the duration of the neomycin selection cells were always maintained at a minimum of $3 \times 10^8$ cells. After 5 days and completion of G418 selection, half of the cells were cultured in ESC medium containing Dox for 3 days and the other half without Dox. GFP-positive cell populations of both Dox-treated and untreated populations were sorted on a FACSAria II cell sorter (BD Biosciences) and flow cytometry data analyzed with FlowJo software. Unsorted mutant populations were served as background controls. Genomic DNA was isolated from GFP-positive sorted and unsorted cells, their sgRNA cassettes amplified by PCR and subjected to NGS sequencing on an Illumina HiSeq 2500. Data analysis was performed as previously described in[32] and gene enrichment determined using MAGeCK[33].

## RNA-seq

$5 \times 10^6$ ESCs were trypsinized and collected by centrifugation. Resulting cell pellets were washed in 1x PBS and resuspended in 1x DNA/RNA protection reagent (Monarch Total RNA Miniprep Kit, NEB). Subsequently, cells were lysed and total RNA extracted following the mammalian cell protocol including optional on-column DNase I treatment.

For RNA-seq library preparation, 1 μg of total RNA per sample was enriched for poly-A using the NEBNext Poly(A) mRNA Magnetic Isolation Module (NEB, E7490) and final RNA-seq libraries generated using the NEBNext Ultra II Directional RNA Library Prep kit (NEB, E7760 and NEBNext Multiplex Oligos) (NEB, E7335/E7500). Final libraries were sequenced as 150 bp paired-end reads on the Illumina HiSeq platform.

## RT-qPCR

For RT−qPCR experiments, total RNA was extracted using the Monarch Total RNA Miniprep Kit (NEB) following the mammalian cell protocol including optional on-column DNase I treatment. 100 ng total RNA was used as input for one-step RT-qPCR using the Luna Universal One-Step RT-qPCR kit (NEB, E3005) on a CFX96 Real-Time PCR System (Bio-Rad). See Supplementary Table 5 for a list of RT-qPCR primers used.

## RNA-seq Data Analysis

Raw paired-end RNA-seq reads were aligned to the mm10 genome using STAR-2.6.1c[70]. Overlap of STAR-aligned reads with genes was performed using HTSeq count function[71] with stranded = reverse option and the GRCm38 version 94 GTF file. The HTseq count matrix was pre-filtered to exclude genes with a read count below 10. Differential gene expression analysis was performed using DESeq2[72] using the "apeglm" method[73] for LFC shrinkage. We applied a threshold of p-adj <0.05 and fold change >0.5 or −0.5 for gene expression changes to be considered significant. Visualization of RNA-seq data was performed using custom R scripts and ggplot2. Gene ontology analysis for significantly deregulated genes was performed using custom R scripts and clusterProfiler refs. [74,75].

## Calibrated ChIP-seq (cChIP) and ChIP-CapSeq

$30 \times 10^6$ ESCs and HEK293T cells were collected, washed once in 1x PBS and crosslinked for 7 min in 1% formaldehyde. The crosslinking was quenched by addition of 125 mM glycine and incubated on ice. The crosslinked cells were pelleted by centrifugation for 5 min at 1200 g at 4 °C. Nuclei were prepared by washes with NP-Rinse buffer 1 (10 mM Tris pH 8.0, 10 mM EDTA pH 8.0, 0.5 mM EGTA, 0.25% Triton X-100) followed by NP-Rinse buffer 2 (10 mM Tris pH 8.0, 1 mM EDTA, 0.5 mM EGTA, 200 mM NaCl). Afterwards, the nuclei were washed twice with shearing buffer (1 mM EDTA pH 8.0, 10 mM Tris-HCl pH 8.0, 0.1% SDS)

and subsequently resuspended in 900 μL shearing buffer with added 1× protease inhibitors complete mini (Roche). Chromatin was sheared by sonication in 15 ml Bioruptor tubes (Diagenode, C01020031) with 437.5 mg sonication beads (Diagenode, C03070001) for 6 cycles (1 min on/1 min off) on a Bioruptor Pico sonicator (Diagenode). For each ChIP reaction 4 % HEK293T-derived human spike-in lysate was combined with ESC lysate and incubated in 1x IP buffer (50 mM HEPES/KOH pH 7.5, 300 mM NaCl, 1 mM EDTA, 1% Triton X-100, 0.1% DOC, 0.1% SDS), with following appropriate antibodies at 4 °C o/n a rotating wheel: H3K27me3 (Diagenode, C15410195), RING1B (Cell Signaling, D22F2), PDS5a (Millipore Sigma #SAB2101764), SUZ12 (Cell Signaling D39F6), H2AK119ub (Cell Signaling D27C4), PCGF1 (Abcam ab202395), RAD21 (Abcam ab992), CTCF (Millipore 070729). Antibody-bound chromatin was captured using Dynabeads protein G beads (Thermofisher #10004D) for 4 h at 4 °C. ChIP washes were performed as described previously[53]. ChIPs were washed 5x with 1x IP buffer (50 mM HEPES/KOH pH 7.5, 300 mM NaCl, I mM EDTA, 1% Triton-X100, 0.1% DOC, 0.1% SDS), or 1.5x IP buffer for H3K27me3 and H2AK119ub, followed by 3x washes with DOC buffer (10 mM Tris pH 8, 0.25 mM LiCl, 1 mM EDTA, 0.5% NP40, 0.5% DOC) and 1x with TE/50 mM NaCl. ChIP DNA was eluted 2x in elution buffer (1% SDS, 0.1 M NaHCO₃) at 65 °C for 20 min, RNase A treated for 30 min at 37 °C, Proteinase K treated for 3 h at 55 °C and crosslinks were reversed o/n at 65 °C. The following day, ChIP samples and corresponding inputs were purified by PCI extraction and DNA precipitation.

### ChIP-qPCR

For ChIP DNA quantification in Extended Fig. 3C qPCR was performed using a CFX Connect Real-Time PCR Detection System (Biorad). See Supplementary Table 6 for a list of qPCR primers used.

### cChIP-seq and ChIPCap-seq library preparation

Libraries were prepared using the NEXTflex ChIP-Seq kit (Bio Scientific) following the "No size-selection cleanup" protocol. Libraries were purified using Agencourt AMPure XP (Beckman Coulter) and amplified using the KAPA Real-Time Library Amplification Kit (KAPABiosystems) following the manufacturer's instructions.

ChIPCap-seq libraries were prepared identically to ChIP-seq libraries. After PCR amplification the libraries were enriched for loci of interest using the MYbaits kit DNA capture target enrichment system (Arbor Biosciences) according to manufacturer's manual. One hundred twenty nucleotides long MYbaits sequence capture probes (Arbor Biosciences) were custom designed against 25 mm9 genomic loci (Supplementary Table 3).

Library quality control including determination of average size and concentration was performed prior to sequencing by commercial Next Generation Sequencing providers. NGS libaries were eventually sequenced as 150 bp paired-end reads on the Illumina HiSeq platform.

### ChIP-seq Data Analysis

Raw reads were mapped to the custom concatenated mouse (mm10) and spike-in human (hg38) genome sequences using bowtie 2 with "–no-mixed" and "no-discordant" options[76]. Subsequently, low quality reads were filtered using SAMtools[77], duplicated reads were discarded with the Picard toolkit (http://broadinstitute.github.io/picard/) and only unique mapped reads were retained.

For visualization uniquely mapped mouse reads were normalized by random subsampling with samtools using calibration factors calculated from the corresponding hg38 spike-in reads as described previously[19,78,79]. High correlation between replicates was confirmed using multiBamSummary and plotCorrelation functions from deepTools[80] before merging for visualization and downstream analysis. Genome coverage tracks (bigWig files) were produced with MACS2' pileup function[81] and heatmaps and profile plots generated with deepTools[80]. Peaks were called on each replicate independently using MACS2[81]. Peaks overlapping with a custom-build blacklist were discarded to remove sequencing artifacts and only peaks called in both replicates were retained for downstream analysis.

### In situ Hi-C

Hi-C was performed as previously described[68] with modifications described in[22] as following: 5 million ESCs were crosslinked in 1% formaldehyde for 10 mins before the reaction was quenched by adding 0.2 M final glycine. Cells were permeabilized in lysis buffer (0.2% IGEPAL, 10 mM Tris-HCl pH 8.0, 10 mM NaCl, 1x Halt Protease inhibitor cocktail) and nuclei isolated in NEBuffer 3 supplemented by 0.3% SDS at 62 °C for 10 min. SDS was quenched with 1% Triton X-100 at 37 °C for 60 min, the nuclei pelleted and resuspended in 250 μl of 1x DpnII buffer with 600 U DpnII (NEB). Following o/n digestion at 37 °C, 200 U were added for 2 h. DpnII was inactivated for 20 min at 65 °C before the DNA ends were filled-in and biotin-marked using Klenow, d(C/G/T)TPs and biotin-14-dATP for 90 min at 37 °C. Proximity ligation was performed using T4 DNA ligase (NEB) for 4 h at room temperature. Subsequently, nuclei were spun down, resuspended in 200 μl mQ water and digested with proteinase K for 30 min at 55 °C in presence of 1% SDS. For crosslink reversal 1.85 M final NaCl was added, and samples incubated at 65 °C o/n. The next day, sample DNA was ethanol precipitated and sheared in 500 μL sonication buffer (50 mM Tris pH 8.0, 0.1% SDS, 10 mM EDTA) on a Bioruptor Pico sonicator (Diagenode). DNA was then concentrated on Amicon ultra 0.5 30 K filter units (Millipore), biotin-pulldown performed using MyOne Streptavidin T1 beads (Life technologies, 65602) and used for NGS library preparation. DNA ends were repaired, biotin removed from unligated ends and NEXTFLEX DNA barcoded adapters (Perkin Elmer) were ligated. Desired PCR cycle numbers were determined in test endpoint PCRs using Q5 DNA polymerase (NEB M0491L). Final HiC libraries were generated from 4–6 individual PCR reactions, which were pooled and subjected to cleanup and size-selection using AMPure beads (Beckman Coulter A63882). Samples were first test sequenced to check library quality before selected Hi-C ibraries were sequenced at greater depth (Supplementary Table 2).

### Hi-C data analysis

Hi-C data were analyzed using the HiC-Pro (2.11.1) pipeline[82]. Read mapping to the mm10 genome was performed using bowtie 2[76] within the HiC-Pro pipeline. PCR and optical duplicates as well as reads with MAPQ < 30 were removed. Filtered valid HiC contact data was binned and raw and ICE normalized.hic contact matrices were generated. We also produced balanced single and multi-resolution.cool and.mcool cooler files for visualization in HiGlass. Virtual 4C tracks were obtained using the hicPlotViewpoint function of HiCExplorer[83–85]. For reference the same analysis was applied to published deep Hi-C data from ESCs[62]. To obtain wild-type ESC TAD and loop information we applied juicer tools[86] "arrowhead" and "HiCCUPS" to the Bonev 2017 ESC dataset. To delineate A and B compartment information, "Eigenvector" function of juicer tools was applied to 250 kb wild-type and Pds5a KO ESC data created in this study. For pileup analysis at HiC loops we used coolpup.py[87] and took averaged the calculated observed over expected interactions within a 105 kb x 105 kb window centered on the loops at 5 kb resolution. For calculation and visualization of compartment strengths, relative contact probabilities, insulation scores and differential Hi-C contact matrices we used the R package GENOVA (https://github.com/robinweide/GENOVA)[88]. Insulation score analysis was conducted as previously described in[89]. Insulation scores[63] were computed at 10 kb resolution using GENOVA. Bins with an insulation score greater than −1 were excluded from the analysis. To qualify as differentially insulated bins, either wild-type or Pds5a KO ESCs had to be lower than −0.2 to exclude very lowly insulated portions of the genome. Insulation gaining and losing regions were defined as bins that had an absolute change in insulation between wild-type and Pds5a KO of 0.2.

### ATAC-seq

ATAC-seq on mESCs was performed in biological triplicates according to protocol as described previously[90,91]. Resulting NGS libraries were sequenced on an Illumina NovaSeq platform in 150 bp paired-end mode.

### ATAC-seq data analysis

ATAC-seq reads were trimmed using NGmerge[92] and aligned to the mm10 genome using bowtie 2 with "–no-mixed" and "no-discordant" options[76]. Low quality reads were filtered using SAMtools[77] and duplicate reads discarded via the Picard toolkit (http://broadinstitute.github.io/picard/).

### ChromHMM chromatin state analysis

Enrichment overlap of RAD21 peaks with different chromatin domains chromatin states was calculated using ChromHMM (version 1.24), a 12-state model based on mouse ENCODE project ChIPseq data[60,93].

### Reporting summary

Further information on research design is available in the Nature Portfolio Reporting Summary linked to this article.

## Data availability

All NGS data reported in this study has been deposited at the Gene Expression Omnibus (GEO) database under accession number GSE194268. Source data are provided with this paper.

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

## Acknowledgements

We are grateful to all members of the Bell, Farnham and Jadhav laboratories, as well as Alexander Stark, Martin Leeb, Jan Michael Peters, Gordana Wutz, Diana Hargreaves, Jesse Dixon, Suhn Rhie, and Geoffrey Fudenberg for feedback and discussions. We especially thank Ilya Flyamer for sharing Hi-C protocols and advice on Hi-C data analysis. We thank Jan Michael Peters and Gordana Wutz for sharing antibodies. We thank Ben Weekley for advice and sharing chromatin fractionation protocols. We thank the Vienna Biocenter Core Facility Next Generation Sequencing. The GMI/IMBA/IMP Scientific Service units and the BioOptics facility. We thank Life Science Editors for editorial assistance. O.B. and U.E. were supported by the Austrian Academy of Sciences. O.B. was supported by the New Frontiers Group of the Austrian Academy of Sciences (NFG-05), the Human Frontiers Science Program Career Development Award (CDA00036/2014-C), and start-up funding from the Norris Comprehensive Cancer Center at Keck School of Medicine of USC.

## Author contributions

D.B., H.F.M., and O.B. initiated and designed the study. D.B., H.F.M. generated cell lines. R.Y. generated parental cell lines. U.E., G.M. provided the CRISPR sgRNA library and helped with the screen design. D.B., H.F.M. performed CRISPR-Cas9 genetic screen. J.W. and G.M. analyzed CRISPR-Cas9 genetic screen data. D.B. performed molecular biology, RNA-seq, ChIP-seq, ATAC-seq and Hi-C experiments. RNA-seq, ChIP-seq, ATAC-seq and Hi-C data analysis was conducted by D.B. O.B. supervised all aspects of the project. The manuscript was prepared by D.B. and O.B. All authors discussed results and commented on the manuscript.

## Competing interests

The authors declare no competing interests.
