## [Peer Review File · Nature Communications]

Loss of Cohesin regulator PDS5A reveals repressive role of Polycomb loopsEditorial Note: This manuscript has been previously reviewed at another journal that is not operating a transparent peer review scheme. This document only contains reviewer comments and rebuttal letters for versions considered at *Nature Communications*.

REVIEWERS' COMMENTS

Reviewer #1 (Remarks to the Author):

The revised version of the manuscript has improved but most of my major concerns have not been addressed. I do appreciate the value of the data presented and I acknowledge that the authors show the relevance of long-range interactions independently of histone modifications for proper silencing of a subset of Polycomb target genes. However, mechanistic understanding on how Pds5A deletion affects cohesin dynamics to alter these ultra long-range interactions is not provided. I would recommend additional changes.

1. Increased cohesin residence on chromatin in Pds5A KO mouse ES cells. This is mentioned several times in the manuscript, but it is not shown.

Authors write in page 12 of the revised MS: "These changes [relative contact reduction at pre-existing wild-type TADs and loops; increased interaction frequencies with neighboring TADs whereas intra-TAD interactions were slightly decreased] resemble those observed upon WAPL and/or PDS5A deletion in cancer cells, where increased cohesin residence time leads to an extension of chromatin loops, resulting in a genome-wide shift towards longer range interactions and violation of TAD boundaries 38,43,45

Ref 38: WAPL KO in HAP1 cells

Ref 43 Wapl KD and Double Pds5A/Pds5B KD in HeLa cells

Ref 45 Pds5A KO in HAP1 cells

In this last ref 45, Van Ruiten et al (2022), the authors include the following statement: "...we found that PDS5A is considerably more abundant than PDS5B (Extended Data Fig. 4e,f). Chromatin looping in HAP1 cells thus appears to be largely controlled by PDS5A".

Thus, in the current manuscript, authors must show if this is also the case in mESC in order to understand the results presented here. Otherwise, they cannot use the data of HAP1 cells (e.g., FRAP data) to sustain their claims. Sentences like the one in line 346

"To uncover the potential mechanism by which increased cohesin residence could interfere with ultra-long Polycomb loops" should be removed/rephrased.

2. - Involvement of other cohesin subunits in Polycomb repression (Rebuttal Fig. 2; revised manuscript Extended Data Fig. 1d)

It is difficult to evaluate the significance of this result if we do not have an idea of the knock down efficiencies. Please provide data of remaining levels of these proteins and of a common cohesin subunit such as Smc1 or Rad21.

3. Page 11 "PDS5A loss mainly affects cohesin occupancy at a subset of target sites."

a. What are these sites in cluster 2? Chromatin state analysis or analysis of histone marks in the different clusters (using public data) would provide information in this regard

b. Can the authors speculate why lack of Pds5A increases occupancy at these particular subset of sites?

c. What is the connection, if any, between these sites of increased cohesin occupancy and sites of increased insulation?

4. Can the authors speculate, based on what is known of cohesin dynamics and the potential role of Pds5 proteins in regulating these dynamics, how ectopic insulation sites affect the ultra long Polycomb interactions?

5. Abstract: "PDS5A deletion leads to aberrant cohesin loop extrusion, ectopic insulation sites and specific reduction of ultra-long Polycomb loops".

Maybe it could be tone down to something like: "PDS5A deletion leads to reduction of ultra-long Polycomb loops and ectopic insulation sites [this is what the data show], possibly through dysregulation of loop extrusion by cohesin [tentative interpretation]"

Reviewer #2 (Remarks to the Author):

The authors have adequately addressed my previous concerns. Specifically, their new experiment where they delete a polycomb site in *Irx2* to achieve derepression of the interacting gene *Foxd1*, is an important addition. Control genes on the same chromosome but without a loop indeed strongly support a role of looping in the repression of polycomb genes. Given this revision, I am supporting the publication of the article.

We thank the reviewers for their thorough consideration of our manuscript. We were pleased that they acknowledged the substantial improvement of our manuscript and thank them for providing insightful feedback and suggestions to further enhance the impact. We have listed the reviewers' comments below and how these were addressed in the revised manuscript. The reviewer comments are in black text and our responses are in blue text.

Reviewer #1:

Comment 1. Increased cohesin residence on chromatin in Pds5A KO mouse ES cells. This is mentioned several times in the manuscript, but it is not shown.

... In this last ref 45, Van Ruiten et al (2022), the authors include the following statement: "...we found that PDS5A is considerably more abundant than PDS5B (Extended Data Fig. 4e,f). Chromatin looping in HAP1 cells thus appears to be largely controlled by PDS5A".

Thus, in the current manuscript, authors must show if this is also the case in mESC in order to understand the results presented here. Otherwise, they cannot use the data of HAP1 cells (e.g., FRAP data) to sustain their claims. Sentences like the one in line 346 "To uncover the potential mechanism by which increased cohesin residence could interfere with ultra-long Polycomb loops" should be remove/rephrased.

Response: We acknowledge that while we do not show bulk increased cohesin association on chromatin, our data clearly demonstrate i) ectopic cohesin binding, ii) formation of new insulation sites, iii) a genome-wide shift towards longer range interactions and iv) violation of TAD boundaries, which together strongly argue that PDS5A is required for regulating cohesin activity by restricting chromatin loop extrusion, in agreement with observations in HAP1 cancer cells (van Ruiten et al., 2022). Nevertheless, addressing the reviewer's comments, we now show in **Extended Data Fig. 4f** of the revised manuscript that *Pds5a* is considerably higher expressed than *Pds5b*, suggesting that it is the dominant paralog. In addition, we have modified the language in the main manuscript including the following sentence: "To uncover the potential mechanism by which aberrant cohesin activity interferes with repressive ultra-long Polycomb loops between PRC1/PRC2 target genes, we defined regions in the genome with significant local changes in 3D chromosome architecture."

Comment 2. Involvement of other cohesin subunits in Polycomb repression (Rebuttal Fig. 2; revised manuscript Extended Data Fig. 1d). It is difficult to evaluate the significance of this result if we do not have an idea of the knock down efficiencies. Please provide data of remaining levels of these proteins and of a common cohesin subunit such as Smc1 or Rad21.

Response: We agree with the reviewer that validation of protein depletion by immune blot would be desirable. However, given the essential role of core cohesin subunits for proliferation and the limited sensitivity of immunoblot detection, quantification of partial protein depletion in the CRISPR mutant populations would be challenging. As an alternative approach, we independently used CRISPR gene editing in TetR-CBX7 reporter ESCs to generate a deletion mutant of *Pds5b*, which is not an essential gene in ESCs. PDS5B depletion was validated by immuno blotting (**Extended Data Fig. 1e, f**). Flow cytometry in the presence of Dox revealed that PDS5B is required for maintenance of CBX7-induced reporter gene silencing, consistent with our conclusion that cohesin regulation by PDS5A and other auxiliary factors is critical for Polycomb function.

Comment 3. Page 11 “PDS5A loss mainly affects cohesin occupancy at a subset of target sites.”

- a. What are these sites in cluster 2? Chromatin state analysis or analysis of histone marks in the different clusters (using public data) would provide information in this regard
- b. Can the authors speculate why lack of Pds5A increases occupancy at these particular subset of sites?
- c. What is the connection, if any, between these sites of increased cohesin occupancy and sites of increased insulation?

Response: We thank the reviewer for suggesting additional data analysis to strengthen our manuscript which we performed and included in the revised version. **a)** We annotated RAD21 peaks in the three clusters using ChromHMM. Cluster 2, which showed significantly increased RAD21 binding, was enriched in proximal and distal gene regulatory sequences and depleted in CTCF sequences compared to clusters 1 and 3 (**Extended Data Fig. 4g**). **b) and c)** In the main text (page 15) and the discussion section (page 18) of the revised manuscript, we speculate that PDS5A deletion causes aberrant chromatin loop extrusion leading to breaching of TAD boundaries, increased cohesin occupancy at gene regulatory elements and formation of ectopic insulation sites. In turn, new insulation sites perturb competing loops mediated by cPRC1.

Comment 4. Can the authors speculate, based on what is known of cohesin dynamics and the potential role of Pds5 proteins in regulating these dynamics, how ectopic insulation sites affect the ultra long Polycomb interactions?

Response: As stated in the revised manuscript on page 16, we speculate that aberrant extension of chromatin loops resulting from reduced cohesin unloading upon PDS5A loss strengthens ectopic cohesin occupancy and interaction domains at the expense of ultra-long Polycomb loops such as between *Dlx2* and the *Hoxd* gene cluster or *Irx2* and *Foxd1*.

Comment 5. Abstract: “PDS5A deletion leads to aberrant cohesin loop extrusion, ectopic insulation sites and specific reduction of ultra-long Polycomb loops”.

Maybe it could be tone down to something like: “PDS5A deletion leads to reduction of ultra-long Polycomb loops and ectopic insulation sites [this is what the data show], possibly through dysregulation of loop extrusion by cohesin [tentative interpretation]”

Response: Based on the recommendation, we modified the abstract as follows: “Instead, PDS5A removal causes aberrant cohesin activity leading to ectopic insulation sites, which disrupt the formation of ultra-long Polycomb loops.”

Reviewer #2:

The authors have dequately addressed my previous concerns. Specifically, their new experiment where they delete a polycomb site in *Irx2* to achieve derepression of the interacting gene *Foxd1*, is an important addition. Control genes on the same chromosome but without a loop indeed strongly support a role of looping in the repression of polycomb genes. Given this revision, I am supporting the publication of the article.

Response: We thank this reviewer for her/his constructive input to our manuscript.